# Heterogeneous Ribonucleoprotein K Is a Host Regulatory Factor of Chikungunya Virus Replication in Astrocytes

**DOI:** 10.3390/v16121918

**Published:** 2024-12-14

**Authors:** Lisa Pieterse, Maranda McDonald, Rachy Abraham, Diane E. Griffin

**Affiliations:** 1W. Harry Feinstone Department of Molecular Microbiology and Immunology, Johns Hopkins Bloomberg School of Public Health, Baltimore, MD 21205, USA; lpieter1@jh.edu (L.P.); dgriffi6@jh.edu (D.E.G.); 2Department of Biochemistry and Molecular Biology, Johns Hopkins Bloomberg School of Public Health, Baltimore, MD 21205, USA; mmcdon47@jh.edu

**Keywords:** alphavirus, astrocytes, chikungunya virus, heterogeneous ribonucleoprotein K, host–virus interactions, neurons, neurovirulence, viral replication, subgenomic RNA translation, RNA-binding protein

## Abstract

Chikungunya virus (CHIKV) is an emerging, mosquito-borne arthritic alphavirus increasingly associated with severe neurological sequelae and long-term morbidity. However, there is limited understanding of the crucial host components involved in CHIKV replicase assembly complex formation, and thus virus replication and virulence-determining factors, within the central nervous system (CNS). Furthermore, the majority of CHIKV CNS studies focus on neuronal infection, even though astrocytes represent the main cerebral target. Heterogeneous ribonucleoprotein K (hnRNP K), an RNA-binding protein involved in RNA splicing, trafficking, and translation, is a regulatory component of alphavirus replicase assembly complexes, but has yet to be thoroughly studied in the context of CHIKV infection. We identified the hnRNP K CHIKV viral RNA (vRNA) binding site via sequence alignment and performed site-directed mutagenesis to generate a mutant, ΔhnRNPK-BS1, with disrupted hnRNPK–vRNA binding, as verified through RNA coimmunoprecipitation and RT-qPCR. CHIKV ΔhnRNPK-BS1 demonstrated hampered replication in both NSC-34 neuronal and C8-D1A astrocytic cultures. In astrocytes, disruption of the hnRNPK–vRNA interaction curtailed viral RNA transcription and shut down subgenomic RNA translation. Our study demonstrates that hnRNP K serves as a crucial RNA-binding host factor that regulates CHIKV replication through the modulation of subgenomic RNA translation.

## 1. Importance

Chikungunya virus (CHIKV) is a mosquito-borne arthritic alphavirus increasingly associated with central nervous system (CNS) infection and subsequent long-term disability. However, information on mechanisms of CHIKV-mediated neurovirulence and neuropathogenesis, especially as it affects astrocytes, the main CHIKV CNS target cell, remains scarce. While viral components of the CHIKV replicase complex have been identified, host protein involvement, and thus elucidation of potential drug targets, remains limited. Our study identified heterogeneous ribonucleoprotein K as a crucial constituent of the CHIKV replicase assembly complex and elucidated its significant regulatory role in viral replication through the modulation of subgenomic RNA translation.

## 2. Introduction

Chikungunya virus (CHIKV) is an emerging mosquito-borne alphavirus that has rapidly spread to Europe and the Americas. Increasingly associated with severe neurological sequelae such as encephalitis and myelitis, or swelling of the brain and spinal cord [1,2,3,4,5,6], respectively, CHIKV infections have mortality rates ranging from 6.1–10.6% [7,8,9], but can result in chronic, disabling manifestations such as general immobility, hand inflexibility, and clinical depression lasting from weeks to years in up to 93.7% of symptomatic cases [10,11,12]. While neonatal [13,14,15], pediatric [16,17,18], and elderly [19,20,21] patients are especially at-risk, no correlations have been identified between sex and age, making neurological CHIKV infection a ubiquitous disease of concern that remains ill-understood. Further confounding the understanding of the CHIKV central nervous system (CNS) pathology includes the scarcity of investigations into the mechanisms behind such CHIKV-mediated neuropathology that would enable development of treatments for this rapidly spreading infectious agent.

While most CHIKV CNS studies direct their attention to the infection of neurons [22,23,24], astrocytes are more often infected [25]. Astrocytes are a major type of glial cell crucial for CNS blood–brain barrier architectural integrity and neurological inflammatory responses [26,27]. Furthermore, astrocytes constitute the most abundant CNS cell type, represent the main CHIKV cellular target for infection of the CNS [25,28,29,30,31,32,33,34,35,36,37], replicate the virus more efficiently than do the neurons, and are directly involved with the neuropathology associated with alphavirus infections [28,30,37,38,39,40].

CHIKV is an 11.8-kb positive-sense, single-stranded RNA virus belonging to the *Togaviridae* family. The 5′-7-methylguanosine capped and 3′-polyadenylated CHIKV genomic RNA contains two open-reading frames (ORFs), as well as a non-coding 3′- and 5′-untranslated region and a non-coding junction [41,42]. Constituting approximately two-thirds of the 5′ genome, the first ORF encodes non-structural proteins (nsPs) 1–4 that comprise the RNA replicase. The 3′ ORF, in contrast, is translated from subgenomic viral mRNA (sgRNA) and encodes six structural proteins, including the capsid (C), envelope 2 (E2), envelope 3 (E3), 6K, TF, and envelope 1 (E1). The CHIKV icosahedral nucleocapsid is enveloped in a lipid bilayer coated with E1 and E2 glycoprotein heterodimers organized in trimers and in a membrane-anchored formation, with E3 mediating the E2 precursor/E1 premature fusion in a low pH-dependent manner [43]. The E1 glycoprotein is a type II fusion protein that mediates viral fusion in a low pH-dependent manner; E2 glycoproteins are type I transmembrane proteins that arbitrate host–receptor interaction [44]. Both the A and B E2 domains contain putative receptor binding sites, with domain B having a PDZ binding motif [39].

The functions of nsP1, nsP2, and nsP4 have been thoroughly investigated. CHIKV nsP1 exhibits methyltransferase, guanylyltransferase, and palmitoylation activities required for RNA capping and host membrane anchoring [45,46]; nsP2 contains an N-terminal helicase region with phosphatase domains responsible for nsP polyprotein processing [47,48,49]. nsP4 is the RNA-dependent RNA polymerase (RdRp) [50]. While the functions of nsP3 remain enigmatic [51,52], nsP3 is critical for efficient CHIKV replication [53]. CHIKV nsP3 is comprised of three domains, including the N-terminus macrodomain that exhibits both mono-ADP-ribose and RNA–binding abilities [54,55], as well as ADP-ribosylhydrolase activities [56]; the alphavirus unique domain (AUD) that binds zinc ions and is involved in nsP3 phosphorylation essential for viral genomic and subgenomic replication [57,58,59]; and, finally, the C-terminal hypervariable domain (HVD), which interacts with host factors necessary for virus replication and host immune evasion, such as Ras-GTPase-activating protein (SH3 domain)-binding protein (G3BP) and the four-and-a-half LIM domain protein 1 (FHL1) [60,61,62,63].

While viral proteins necessary for alphavirus RNA initiation complex formation have been identified, host proteins involved in this late RNA synthetic process remain enigmatic and uncharacterized. Previous alphavirus cross-link assisted messenger ribonucleoprotein (CLAMP) and coimmunoprecipitation studies have elucidated various members of the heterogeneous nuclear ribonucleoproteins (hnRNPs) subfamily, a complex of nuclear RNA-binding proteins involved with RNA processing, trafficking, and translation, as potential host proteins associated with late replicase complex components [64,65,66,67,68,69]. One such member, hnRNP K, is enriched in human embryonic kidney (HEK) 293 cells infected with the prototype alphavirus, Sindbis virus (SINV) [70], and has similarly been identified as an augmented component in BHK-21 cytoplasmic membrane fractions opulent in late SINV replicase complex assembly constituents, including nsP1-3 [71]. Further investigations involving coimmunoprecipitation have elucidated a SINV viral RNA-binding site directly interacting with hnRNP K [69]; furthermore, the site-directed mutagenesis of such an hnRNPK–vRNA binding site significantly hampers SINV viral replication kinetics, assumedly due to disrupted hnRNPK–vRNA interaction, and thus subsequent interference of hnRNP K interaction with viral non-structural proteins and sgRNA, but not genomic RNA (gRNA) [71]. Additionally, RNAi-mediated knockdown of host protein hnRNP K also significantly restricts SINV virus replication in vitro [67,68]. HnRNP K colocalization with late RNA replicase complex components, as well as the ensuing curtailed viral replication kinetics associated with disrupted hnRNPK–vRNA interaction, indicates that hnRNP K may represent a key host protein player involved in late replication complex activities and, thus, replication regulation. However, the mechanistic role hnRNP K plays as a potential host component of the late alphavirus replication initiation complex remains enigmatic. Moreover, investigations into hnRNPK–vRNA interaction in CHIKV, specifically, have yet to be conducted. Preliminary studies performed in murine neuronal (NSC-34) and astrocytic (C8-D1A) cell lines interestingly indicated that hnRNP K appears to be modified during CHIKV infection in astrocytes, but not neurons, in vitro. Considering cell type-specific differences in CHIKV infection dynamics [25], as previously discussed, we strived to determine whether hnRNP K represents a crucial host component of the CHIKV late replicase complex in astrocytes and neurons and to illuminate differences in hnRNP K interaction within these two cell types. We hypothesized that hnRNP K plays a prominent role as a regulator of viral sgRNA and, thus, sgRNA translation, in both astrocytes and neurons in vitro, but that hnRNP K plays a novel role in sgRNA translation within astrocytes, the main CNS target.

## 3. Materials and Methods 

### 3.1. HnRNPK–vRNA Site-Directed Mutagenesis

Site-directed mutagenesis was performed using CHIKV 181/25 template plasmid DNA (pDNA), the primers delineated in Table 1, and Pfu Turbo DNA polymerase (Stratagene, La Jolla, CA, USA; Cat. No. 600250-52), according to manufacturer protocols. The amplified product was purified using a QIAquick PCR Purification Kit (Qiagen, Hilden, Germany; Cat. No. 28104) and treated with DpnI enzyme (New England BioLabs, Ipswich, MA, USA; Cat. No. R0176S) to remove the methylated and hemi-methylated pDNA. The resulting purified product was then transformed into Max Efficiency DH5α Competent Cells (Invitrogen, Waltham, MA, USA; Cat. No. 18258012) and plated on selective media according to manufacturer protocols. The transformants were selected, sequenced, and expanded into selective liquid media. Plasmid isolation using a QiaPrep Midi Kit (Qiagen, Cat. No. 12143) was performed the following day.

### 3.2. CHIKV Mutant Virus Preparation from Full-Length Plasmid

Plasmid was linearized via NotI (New England BioLabs, Cat. No. R0189S) restriction digestion, according to manufacturer protocols. Digest efficiency verification via gel electrophoresis, phenol chloroform extraction of plasmid, and precipitation using ethanol and sodium acetate was performed. After pellet dehydration, the plasmid pellet was resuspended in nuclease-free water and quantified via a NanoDrop Spectrophotometer ND-1000 (Thermo Fisher Scientific, Waltham, MA, USA). The synthesis of mRNA from purified, linearized plasmid DNA was performed using an mMESSAGE mMACHINE SP6 Transcription Kit (Invitrogen, Cat. No. AM1340), according to manufacturer protocols. The remaining plasmid DNA was digested using TURBO DNase, and the resulting product was transfected into confluent BHK-21 fibroblasts (ATCC, CCL-10) using Lipofectamine 2000 Transfection Reagent (Invitrogen, Cat. No. 11668019) according to manufacturer protocols. Media was then aspirated and replaced with 12 mL DMEM ([+]4.5 g/L D-glucose, [+]L-glutamine, [−] sodium pyruvate; Thermo Fisher Scientific, Cat. No. 11965-092), supplemented with 10% heat-inactivated FBS (Thermo Fisher Scientific, Cat. No. 26140-079), 1% penicillin-streptomycin (Thermo Fisher Scientific, Cat. No. 15140-122), and 2 mM L-glutamine (Thermo Fisher Scientific, Cat. No. 25030-081), and the flask was allowed to incubate for 24 h at 37 °C/5% CO_2_. Following observation of cytopathogenic effects, the supernatant of the lysed cells was collected, filtered, and aliquoted for storage. This stock, designated as “passage zero” (P0), was used to infect a low-passage, confluent T150 flask of BHK-21 fibroblasts, according to previously delineated protocols. The resulting virus, designated as “passage 1” (P1), was collected and titered via Vero plaque assays.

### 3.3. In Vitro P1 Virus Infections

Media from low passage, confluent hybridized mouse motor neurons (NSC-34, a kind gift from Neil Cashman) and mouse astrocytes (C8-D1A, ATCC, CRL-2541) plated on uncoated 6-well plates (Corning Inc., Corning, NY, USA; Cat. No. 3506) was aspirated and replaced with 200 µL of P1 inoculum prepared at a multiplicity of infection (MOI) 1:5 (cells–virus), according to previously described protocols [25] using DMEM supplemented with 1% FBS, 1% penicillin-streptomycin, and 2 mM L-glutamine as diluent. Supernatant, RNA, and protein specimens were collected at designated timepoints, as described below.

### 3.4. Vero Cells Plaque Assays

Supernatant samples collected from infected cells were serially diluted using DMEM supplemented with 1% heat-inactivated fetal bovine serum (FBS, Thermo Fisher Scientific, Cat. No. 26140-079), 1% penicillin-streptomycin (Thermo Fisher Scientific, Cat. No. 15140-122), and 2 mM L-glutamine (Thermo Fisher Scientific, Cat. No. 25030-081). Media from low passage, confluent Vero (ATCC, CCL-81) cell cultures, plated on 12-well plates (Corning Inc., Cat. No. 3512), was aspirated and serially diluted inoculums were added to designated wells, incubated, and overlaid with bacto agar solution, according to previously described protocols [25,54]. The plates were incubated for 48 h in a designated 37 °C/5% CO_2_ incubator. Fixation and staining were performed using 10% formaldehyde (Thermo Fisher Scientific, Hampton, U.S.A; Cat. No. F79-1) and 0.02% *w*/*v* crystal violet ethanol solution, according to previously published protocols [25,54]. With the aid of a transilluminator, plaque-forming units (PFUs) were manually counted and normalized to both the dilution factor and the inoculum volume; PFU per mL = (Plaques Counted × Dilution Factor)/(Inoculum Volume). Biological replicates were collected and processed for each time-point investigated.

### 3.5. RNA Extraction and qPCR

Infected cell monolayers were washed once using 2 mL 1× PBS. Cell lysates were then prepared via the addition of 700 µL of Buffer RLT (Qiagen, Cat. No. 79216) to each monolayer, which were mechanically disrupted using a cell scraper (Sarstedt, Nümbrecht, Germany; Cat. No. 83.3951). The resulting cell lysates were then processed using RNeasy Mini Kits (Qiagen, Cat. No. 74106) according to manufacturer protocols. RNA quantity and quality was verified via NanoDrop spectrophotometry. Employing 1 μg total RNA per reaction, the synthesis of cDNA was performed using a SuperScript III First-Strand Synthesis System (Invitrogen, Cat. No. 18080051) kit according to manufacturer protocols. RT-qPCR reactions were prepared using GoTaq qPCR Master Mix (Promega Corporation, Madison, NY, USA; Cat. No. A600A), 0.4 µM forward primer, 0.4 µM reverse primer (IDT, Newark, U.S.A; Table 1), and 3 µL of 1:10 diluted cDNA per reaction, according to manufacturer protocols. Cycle threshold (Ct) values were standardized to plasmid copy numbers, which were normalized to *Gapdh* (Applied Biosystems, Waltham, MA, USA; Cat. No. 4308313) copy numbers, according to previously described protocols [54].

### 3.6. Infectious Center Assay with Virus

C8-D1A and NSC-34 cells were infected with attenuated CHIKV 181/25 or ΔhnRNPK-BS1 at an MOI of 5 and incubated for 1 h at 4 °C and then at 37 °C for another hour. Following cell trypsinization, ten-fold serial dilutions of cells in DMEM supplemented with 1% heat-inactivated FBS were prepared, counted, and plated on confluent Vero cells. Cells were then directly overlaid with 1.5 mL of 0.6% bacto agar in 1× MEM (Thermo Fisher Scientific, Cat. No. 11935046) supplemented with 2% FBS. The plates were incubated at 37 °C for 48 h, fixed with 10% formaldehyde prepared in PBS, and stained with 0.02% crystal violet ethanol solution. The plaques were then counted and normalized to the total cell number, as determined via trypan blue exclusion. The values are represented as infectious centers per 10^5^ cells/mL.

### 3.7. Western Blots

Following a PBS washing step, cell monolayers were lysed using 100 µL of cold RIPA buffer (10 mM Tris-HCl, pH 9.0, Fisher Scientific, BP152-1; 1 mM EDTA, Fisher, BP120-500; 1% Triton X-100, J.T. Baker, Phillipsburg, NJ, USA, Cat. No. X198-07; 0.1% sodium deoxycholate, Sigma-Aldrich, St. Louis, MO, USA, Cat. No. D6750-100G; 1% SDS, Bio-Rad, 161-0302; 140 mM NaCl, Fisher, BP358-212; prepared in Milli-Q water). Sample protein concentrations were quantified and normalized using a commercial colorimetric assay (Bio-Rad, Hercules, CA, USA; Cat. No. 5000111). The addition of 6× cracking buffer (0.35 M Tris pH 6.8, 30% glycerol, 10% SDS, 0.125% bromophenol blue, 0.05% 2-mercaptoethanol) to each sample, followed by a 10 min heat-block boiling step, was performed. Samples were then run on an SDS-PAGE gel consisting of a separating gel (10% acrylamide/bis solution, 29:1, Bio-Rad, Cat. No. 1610156; 3.76 M Tris; 0.1% SDS; 0.1% APS; 0.1% TEMED) and a stacking gel solution (5.12% acrylamide/bis solution, 29:1, Bio-Rad, Cat. No. 610156; 0.13 M Tris; 0.1% SDS; 0.1% APS; 0.1% TEMED), while nitrocellulose membrane transfer, blocking, and antibody incubations were performed, according to previously described protocols [25].

### 3.8. Nuclear and Cytoplasmic Fractionation

Protein lysates were collected from infected cells, according to protocols delineated in Section 3.7. Nuclear and cytoplasmic fractionation was performed using an NE-PER Nuclear and Cytoplasmic Extraction Reagents kit (Thermo Fisher Scientific, Cat. No. 78833) according to modified manufacturer protocols that included three additional cold PBS washes of the nuclear pellet to minimize cytoplasmic contamination.

### 3.9. RNA Coimmunoprecipitation

Low passage, confluent C8-D1A and NSC-34 monolayers were infected using CHIKV 181/25 P1, CHIKV ΔhnRNPK-BS P1, or media (mock) at an MOI of 1:5 (cells–virus), according to previously described protocols (see Section 3.3). At designated hours post-infection, designated wells were washed once using 2 mL 1× PBS and lysed using 700 µL per well of the following in-house lysis buffer: 50 mM HEPES pH 7.4, 150 mM NaCl, 1 mM MgCl_2_, 1 mM EGTA, 1% Triton X-100, 1 mM DTT, 1 mM NaF, and 1× cOmplete Mini Protease Inhibitor Cocktail (Roche, Basel, Switzerland; Cat. No. 11836153001). Mouse IgG2a anti-hnRNP K monoclonal antibody or a mouse IgG2a isotype control (Table 2) was added to each lysate at a final concentration of 1:250. Half of the samples were processed without the addition of any antibodies, serving as input controls. The antibody-treated samples were incubated at 4 °C overnight and then incubated with 35 µL of resuspended Pierce Protein A/G Agarose beads (Thermo Scientific, Cat. No. 20421) for 5 h at room temperature. Resulting agarose pellets were washed thrice using the aforementioned RNA co-IP lysis buffer, eluted using an in-house elution buffer (2% 2-mercaptoethanol, 1× SDS), and boiled for 10 min at 95 °C in a heating block to permit the complete dissociation of the nucleoprotein complexes. RNA isolation was performed via the addition of chloroform and centrifugation for 15 min at 12,000× *g* at 4 °C. The resulting aqueous phase was precipitated using 100% isopropanol, and the ensuing RNA pellet was washed with 75% ethanol. Following pellet air drying and resuspension, cDNA synthesis was performed using a SuperScript III First-Strand Synthesis System (Invitrogen, Cat. No. 18080051) kit according to manufacturer protocols. RT-qPCR was carried out according to previously described protocols (see Section 3.5) using primer sets delineated in Table 1. The resulting Ct values were standardized to the plasmid copy numbers and normalized to the input controls. The comparison of the experimental antibody to the isotype control was performed to ensure binding specificity.

### 3.10. Protein Coimmunoprecipitation

Low-passage, confluent C8-D1A and NSC-34 cells were infected using CHIKV 181/25 P1, CHIKV ΔhnRNPK-BS1 P1, or mock at an MOI of 1:5 (cells–virus), according to previously described protocols (see Section 3.3). At select hours post-infection, designated wells were washed once using 2 mL 1× PBS and lysed using 100 µL per well of an in-house RIPA co-IP lysis buffer (50 mM Tris pH 8, 150 mM NaCl, 1% NP40, 5% Na deoxycholate, 1 mM EDTA) supplemented with cOmplete Mini Protease Inhibitor Cocktail (Roche, Cat. No. 11836153001). Mouse IgG2a anti-hnRNP K monoclonal antibody or a mouse IgG2a isotype control (Table 2) was added to each lysate at a final concentration of 1:250 and allowed to incubate overnight at 4 °C on a rotating shaker. The next day, 35 µL of Pierce Protein A/G Agarose beads was added to each sample, incubated for 5 h at room temperature, and washed thrice using 500 µL of lysis buffer. The samples were then processed using 6× SDS cracking and analyzed via Western blot analysis, according to previously delineated protocols (see Section 3.7).

### 3.11. siRNA Transfections

Media from low-passage, confluent C8-D1A cells were replaced 1 h prior to transfection. Lipofectamine-mediated transfection of C8-D1A cells with commercial siRNA constructs mm.hnRNPK.Ri.13.1, mm.hnRNPK.Ri.13.2, mm.hnRNPK.Ri.13.1 (Kit #498320079, IDT), or an *Hrpt* control was performed. One construct, mm.hnRNPK.Ri.13.3, showcased significantly downregulated relative *Hnrnpk* expression (Appendix A) and was thus selected as the primary siHnRNPK construct. Transfection using a 100 nM construct per well was performed using Lipofectamine 3000 Transfection Reagent (Thermo Fisher, Cat. No. L3000001) according to manufacturer protocols. The transfected cells were incubated for 48 h in a designated 37 °C/5% CO_2_ chamber prior to CHIKV infection. The collection and isolation of RNA, followed by cDNA synthesis and RT-qPCR, was performed to analyze *Hnrnpk* gene expression normalized to *Gapdh* (Thermo Fisher, Cat. No. 4308313).

### 3.12. shRNA Transfections and Transductions

Constitutive C8-D1A hnRNP K downregulation was performed via co-transfection of 3 µg pMD2.G envelope vector; 6 µg psPAX2 plasmid encoding gag, pol, rev, tat, and accessory proteins; and 10 µg of a commercially available short hairpin RNA (shRNA) transfer vector (TRCN0000096825, Millipore Sigma), generating siRNAs that mediate hnRNP K gene expression interference. An additional lentiviral construct encoding the empty shRNA backbone (pLKO.1—TRC, AddGene Plasmid #10878) was used as a control. Transfection was performed using Lipofectamine 3000 Transfection Reagent (Thermo Fisher, Cat. No. L3000001) according to manufacturer protocols. Transfected HEK-293 cells were incubated for 48 h at 37 °C/5% CO_2_. The resulting lentiviral supernatants were then collected, and the C8-D1A cells were transduced for 48 h, followed by 2 μg/mL puromycin selection for 72 h. Individual cells were plated and expanded to ensure clonal homogeneity. The downregulation of hnRNP K was confirmed via both RT-qPCR and immunoblotting.

## 4. Results

### 4.1. Mutagenesis of HnRNPK–vRNA Modifies vRNA Secondary Structure

Genomic cross-sequence alignment to the SINV hnRNPK–vRNA reference binding site [69] illustrated conservation of the same binding site in CHIKV, which was identified to be 5′-GCCATAGTTTTAGGAGGAGCTAAT-3′. Coimmunoprecipitation experiments involving the RT-qPCR amplification of cross-linked RNA molecules from antibody-mediated pulled-down hnRNP K protein confirmed the amplification of the hnRNPK–vRNA-binding site sequence in CHIKV-infected, but not uninfected, astrocytes (Figure 1A). To investigate the disruption of the hnRNPK–vRNA interaction, we inserted four (ΔhnRNPK-BS1) or eight (ΔhnRNPK-BS2) silent mutations via site-directed mutagenesis-mediated base-pair substitution into the postulated hnRNPK–vRNA-binding site (“BS”; Figure 1B,C), resulting in CHIKV ΔhnRNPK mutants encoding hnRNPK–vRNA-binding sites with unaltered translation products, yet changes in RNA secondary structure. Modeling of this mutated viral RNA-binding site using RNAStructure [72], RNAComposer [73], and Geneious software illustrated dramatically modified RNA secondary structure characteristics, including an expanded hairpin loop structure, the addition of an internal loop, and a disrupted helix formation, as well as the exclusion of the 5′ dangling end in both the ΔhnRNPK-BS1 and ΔhnRNPK-BS2 mutants (Figure 1D–I). RPISeq [74] RNA–protein interaction software, using RNA sequence information, illustrated the diminished probability of murine host hnRNP K interaction with the ΔhnRNPK-BS1 hnRNPK–vRNA sequence (RPISeq-RandomForest[RF]_BS1_ = 0.35; RPISeq-SVM_BS1_ = 0.79) versus 181/25 (RPISeq-RF_181/25_ = 0.45; RPISeq-SVM_181/25_ = 0.83). Interestingly, the ΔhnRNPK-BS2 mutants showcased a lower random forest, but identical support vector machine (SVM), probability of hnRNPK–vRNA interaction relative to 181/25 (RPISeq-RF_BS2_ = 0.35; RPISeq-SVM_BS2_ = 0.83), indicating increased murine hnRNP K hnRNPK–vRNA-binding affinity to the ΔhnRNPK-BS2 binding site than to that of ΔhnRNPK-BS1, presumably due to the ΔhnRNPK-BS2 RNA secondary structures more closely mimicking those of the 181/25 consensus sequence.

### 4.2. Disruption of hnRNPK–vRNA Negatively Regulates CHIKV Replication

The infection of murine C8-D1A astrocytes and NSC-34 neurons with the synthesized CHIKV ΔhnRNPK-BS1 and ΔhnRNPK-BS2 mutants significantly hampered ΔhnRNPK-BS1 and ΔhnRNPK-BS2 viral replication in both cell lines (Figure 2A,B), as quantified by Vero plaque assays. However, the viral replication kinetics of ΔhnRNPK-BS1 appeared to be more profound, presumably due to secondary structure modifications associated with diminished hnRNPK–vRNA binding, as previously discussed. Thus, we proceeded to utilize the ΔhnRNPK-BS1 mutant as a model for characterizing hnRNPK–vRNA binding. Infectious center assays used to quantify intracellular viral titers following 1 h of virus exposure at an MOI of 5 demonstrated significantly enhanced CHIKV 181/25 viral titers in astrocytes, albeit a nonsignificant decrease in neuronal cultures, as well (Figure 2C). As expected, CHIKV ΔhnRNPK-BS1 virions initiated viral replicase assembly, as quantified by the infectious centers, more effectively in astrocytes (C8-D1A) versus neurons (NSC-34), indicating that hnRNP K potentially serves as a host factor involved in CHIKV replicase assembly or composition in both cell types. Due to such conservation of the ΔhnRNPK-BS1 phenotype within both astrocytes and neurons, future experiments were predominantly performed using C8-D1A, an astrocyte cell line derived from mouse cerebellum and previously utilized as a model for astrocyte CHIKV infection [25].

The disruption of the hnRNPK–vRNA interaction was additionally experimentally confirmed via anti-hnRNP K or antibody isotype control (IgG2a) coimmunoprecipitation experiments involving hnRNP K pull-down from CHIKV 181/25 or CHIKV ΔhnRNPK-BS1-infected C8-D1A protein lysates, followed by isolation of hnRNP K-bound total RNA, cDNA synthesis, and RT-qPCR to quantify CHIKV hnRNPK–vRNA sequence quantities. The results illustrated significantly diminished quantities of the hnRNPK–vRNA-binding site sequences in the ΔhnRNPK-BS1 mutant (Figure 2D), thus indicating decreased astrocytic hnRNP K protein interaction with the identified vRNA-binding site sequence. Together, these results verified the interaction disruption of host hnRNP K with the hypothesized viral RNA-binding site in our ΔhnRNPK-BS1 model.

### 4.3. HnRNP K Interacts with Viral Subgenomic RNA and Regulates CHIKV Subgenomic RNA Translation via HnRNPK–vRNA-Binding

Whether the hnRNPK–vRNA interaction acts as a determinant of downstream hnRNPK–sgRNA-binding in CHIKV models remains unknown. Considering previous SINV studies illustrating hnRNP K interaction with alphavirus sgRNA, as well as the hnRNPK-mediated regulation of SINV sgRNA translatability [69], we postulated that the decreased CHIKV fitness observed in the ΔhnRNPK-BS1 mutants may be due to disruption of the hnRNP K interactivity with sgRNA, an interaction postulated to be essential for sgRNA translation. Therefore, to investigate whether modification of the hnRNPK–vRNA interaction decreases the propensity of hnRNP K to interact with viral sgRNA, we performed hnRNP K antibody-mediated coimmunoprecipitation experiments by pulling-down hnRNP K or antibody isotype control protein from CHIKV 181/25, ΔhnRNPK-BS1, or mock-infected C8-D1A murine astrocytes at 24 h post-infection (HPI). RNA isolation, cDNA synthesis, and RT-qPCR of the pulled-down product were performed to quantify viral sgRNA expression. Modification of the hnRNPK–vRNA-binding site was associated with modest, yet significant, decreases in astrocytic hnRNP K interaction with sgRNA (Figure 3A), indicating that hnRNPK–vRNA interaction disruption appears to act as at least a minor determinant of hnRNPK–sgRNA-binding propensity.

To elucidate hnRNP K interactions with CHIKV RNA species, we quantified genomic and subgenomic RNA transcripts in total RNA isolated from CHIKV 181/25 or ΔhnRNPK-BS1-infected astrocyte cells at 24 HPI. RT-qPCR analysis using synthesized cDNA illustrated dramatically diminished sgRNA transcription, as well as a modest decrease in gRNA synthesis (Figure 3B,C). Similarly, Western blot analysis demonstrated a decrease in both structural and non-structural protein production in CHIKV ΔhnRNPK-BS1 mutant-infected astrocytes; however, the shut-down of structural glycoproteins appears to be more prominent, indicating the impertinent role of hnRNP K, especially in sgRNA translation (Figure 3D–F). Furthermore, hnRNP K pull-down experiments indicated no hnRNP K coimmunoprecipitation with E2 glycoprotein, suggesting hnRNP K regulation via sgRNA interaction directly. Immunoblotting and densitometric analyses of nuclear and cytoplasmic protein fractions confirmed hnRNP K cytoplasmic translocation during astrocytic CHIKV infection (Figure 3H,I). Densitometric analyses additionally demonstrated a slight upregulation in the hnRNP K cytoplasmic signal in ΔhnRNPK-BS1- versus 181/25-infected C8-D1A cells over time, indicating increased hnRNP K translocation in the hnRNPK–vRNA disrupted mutants.

Considering the hnRNPK–vRNA disruption-mediated downregulation of both CHIKV sgRNA and gRNA (Figure 3B,C), we performed preliminary investigations regarding the hnRNP K stabilization of both RNA species. Infection of C8-D1A cells with CHIKV 181/25 or ΔhnRNPK-BS1 was performed for 1 h, followed by CHIKV nsP4 (RdRp) inhibition via sofosbuvir treatment [75]. RNA lysates were collected at 12 HPI, and RT-qPCR analysis using isolated total RNA was performed to examine the quantities of both sgRNA and gRNA. RdRp inhibition significantly decreased both RNA species in 181/25 and ΔhnRNPK-BS1; however, no sgRNA or gRNA was detected in sofosbuvir-treated ΔhnRNPK-BS1, suggesting that the hnRNP K stabilization of the vRNA components may be one mechanism explaining differential vRNA transcriptional profiles between 181/25 and ΔhnRNPK-BS1 (Figure 3J,K).

### 4.4. Host hnRNP K Availability Regulates CHIKV Replication in Astrocytes

To determine whether hnRNP K availability itself regulates CHIKV replication in astrocytes, we performed *Hnrnpk* transcript knockdown via small interfering RNA (siRNA) transfection using a commercially verified hnRNP K-specific knockdown siRNA sequence and a non-targeting control, the former downregulating *Hnrnpk* transcript levels by approximately 80% in transfected cells versus negative controls (Appendix A). The continuous downregulation of the *Hnrnpk* transcripts was furthermore confirmed via RT-qPCR analysis of the RNA lysates collected from siHnRNPK-transfected, CHIKV 181/25-infected astrocytes at various time-points (Figure 4A). Following such quality control, Vero plaque assays demonstrated significantly diminished longitudinal CHIKV 181/25 viral titers in siHnRNPK-transfected C8-D1A astrocytes (Figure 4B). As expected, hnRNP K downregulation had no effect on ΔhnRNPK-BS1 titers (Figure 4C). Importantly, the CHIKV 181/25 and ΔhnRNPK-BS1 titers were identical in siHnRNPK-transfected astrocytes (Figure 4D), indicating that site-directed mutagenesis of the hnRNPK–vRNA-binding site yields similar regulatory consequences as those caused by decreased hnRNP K availability in the 181/25-infected astrocytes. siHnRNPK-transfection furthermore dramatically ablated transcription of both 181/25 sgRNA and gRNA; however, such a decrease in transcript levels, while significant at 36 HPI, was only modestly observed in ΔhnRNPK-BS1-infected C8-D1A cells (Figure 4E,F), as quantified via RT-qPCR. Furthermore, hnRNP K availability appears to only modestly positively regulate infectious center formation, suggesting that the hnRNP K modulation of sgRNA, via recruitment of translational machinery or vRNA stabilization and decay avoidance, may be the main role of hnRNP K in CHIKV replication (Figure 4G).

Constitutive C8-D1A hnRNP K downregulation was performed via the co-transfection of pMD2.G envelope vector; psPAX2 plasmid encoding gag, pol, rev, tat, and accessory proteins; and a commercially available short hairpin RNA (shRNA) transfer vector generating siRNAs mediating hnRNP K gene expression interference. An additional lentiviral construct encoding the empty shRNA backbone was used as a control. Following plasmid co-transfection in HEK-293 cells to generate lentiviral constructs, C8-D1A transduction and puromycin selection were performed to generate shHnRNPK and shControl C8-D1A clones. The clones were then infected with CHIKV 181/25 or ΔhnRNPK-BS1, and *Hnrnpk* downregulation was confirmed via RT-qPCR analysis at 12 HPI (Figure 4H). Vero plaque assays demonstrated significantly diminished 181/25 (Figure 4I), but not ΔhnRNPK-BS1 (Figure 4J), titers in the shHnRNPK C8-D1A clones. However, the 181/25 viral titers remained higher relative to those of ΔhnRNPK-BS1 in both the shHnRNPK and shControl clones (Figure 4K,L). RT-qPCR analysis illustrated significantly diminished sgRNA and gRNA transcripts in cells with constitutive hnRNP K downregulation (Figure 4M,N).

### 4.5. Host HnRNP K Availability Regulates Viral Structural Protein Synthesis

In order to determine whether the levels of host hnRNP K regulate viral protein synthesis, we downregulated hnRNP K expression via the transient or constitutive systems previously discussed. Cell lysates were collected and hnRNP K expression was determined through Western blotting and densitometric analysis. Firstly, transient siRNA-mediated hnRNP K protein downregulation was foremost confirmed via immunoblotting (Figure 5A,B). Densitometric analyses unexpectedly demonstrated significantly increased E2 glycoprotein expression in siHnRNPK-transfected C8-D1A astrocytes infected with CHIKV 181/25, but not ΔhnRNPK-BS1 (Figure 5C), which contrasts with the diminished sgRNA transcription observed (Figure 4E,F). siHnRNPK downregulation resulted in no dramatic changes in nsP2 expression (Figure 5D), indicating hnRNP K involvement with sgRNA interaction, modulation, and translational regulation. Secondly, constitutive hnRNP K downregulation via the generation of C8-D1A shHnRNPK clones with diminished hnRNP K expression similarly demonstrated increased E2 glycoprotein synthesis upon CHIKV 181/25 and ΔhnRNPK-BS1 infection (Figure 5E–G), which similarly contrasts with the transcriptional profiles observed within the same cell clones (Figure 4M,N). As observed in the siHnRNPK transient downregulation experiments, nsP2 protein expression did not change upon shHnRNPK constitutive downregulation (Figure 5H). Thus, both transient and constitutive hnRNP K downregulation in CHIKV-infected astrocytes demonstrate the importance of hnRNP K availability in subgenomic protein translation.

## 5. Discussion

Previous SINV studies have identified hnRNP K as a host factor associated with the SINV replicase complex [69,71]. However, no investigation into hnRNP K involvement in CHIKV replication has been undertaken, and no studies have examined the hnRNP K mechanisms in CHIKV-infected astrocytes, the main CNS cellular target of CHIKV. Our studies provide the first primary evidence of hnRNP K host factor involvement in both the early and late CHIKV replicase complex in infected astrocytic and neuronal cells in vitro. We first examined the hnRNP K interaction with the predicted viral RNA-binding site based upon a previous SINV model [69] and then developed a CHIKV hnRNPK–vRNA-binding site mutant, ΔhnRNPK-BS1, with diminished hnRNPK–vRNA interaction, as verified via coimmunoprecipitation experiments and bioinformatical modeling. Using this model, we demonstrated that disruption of the hnRNPK–vRNA interaction is associated with significant CHIKV attenuation in both murine C8-D1A astrocytes and NSC-34 neurons in vitro, suggesting that hnRNP K is a conserved host factor involved in CHIKV replication, regardless of non-neuronal or neuronal cell type. To elucidate the mechanism behind such viral attenuation, we performed infectious center assay experiments and found significantly diminished CHIKV ΔhnRNPK-BS1 mutant infectivity in astrocytes, which provided valuable preliminary evidence of hnRNP K involvement in replication initiation complex formation. Therefore, we postulated that the hnRNPK–vRNA interaction may play roles in vRNA stability, transcription, or translation as part of the replication complex assembly, since the hnRNPK–vRNA-binding site is assigned to an open reading frame encoding structural proteins. The quantification of viral RNA species in CHIKV-infected astrocytes via RT-qPCR demonstrated significantly diminished gRNA and sgRNA expression within astrocytes infected with the hnRNPK–vRNA mutant, suggesting hnRNPK–vRNA regulatory or synthetic involvement in CHIKV transcription. However, decreased gRNA may be merely attributed to diminished viral titers associated with the hnRNPK–vRNA disruption phenotype, as hnRNP K is known to interact with sgRNA only [71]. Furthermore, hnRNP K transcript and subsequent early protein knockdown via transient siRNA transfection or constitutive shRNA-mediated hnRNP K downregulation was associated with the significantly diminished transcription of both gRNA and sgRNA, the phenotype of the former postulated to be explained by decreased viral titers. HnRNP K is a known transcription factor modulating dengue virus, human papillomavirus, hepatitis B virus, and hepatitis C virus [76]. Observations of changes in both gRNA and sgRNA species under conditions of hnRNP K downregulation indicate that hnRNP K may be a regulatory transcription factor in both the early and late replicase assembly, which would need to be verified through further investigation into RNA fluorescence in situ hybridization and hnRNP K colocalization studies in the context of 181/25 versus ΔhnRNPK-BS1 infection. Furthermore, while viral components of the early and late replication complex have been characterized, the host proteins involved remain ill-understood. Investigations into the CHIKV-mediated cytoplasmic spherule formations containing replication complex constituents identified dense condensates situated close to the spherule neck structures [77]; further cryogenic electron microscopy studies illustrated that the crown architectural component of these spherules consists of polymerized nsP3 protein that recruits host factors via its HVD domain [78]. However, nsP3-rich spherule conglomerates, and thus host factor constituents colocalizing within CHIKV, remain to be identified. Future research into longitudinal hnRNP K coimmunoprecipitation with early and late replication formations is needed to elucidate the transcriptional activities of hnRNP K.

Interestingly, the translation of sgRNA transcripts was found to be severely ablated upon hnRNPK–vRNA disruption, but not by hnRNP K knockdown, suggesting that hnRNP K, in addition to performing transcriptional regulatory roles, directly modulates the translation of structural proteins encoded by sgRNA through direct interaction with vRNA and sgRNA, as previously observed by others [79]. Thus, diminished hnRNP K availability alone is not associated with halted hnRNPK–vRNA-binding affinity, thus explaining the lack of diminished E2 glycoprotein synthesis. To further delineate this E2 translation phenotype observed within the hnRNP K knockdown astrocyte clones, electrophoretic mobility shift and fluorescent polarization assays need to be performed to determine the hnRNPK–vRNA dissociation constant relative to the hnRNP K interaction with host RNA, as viral-host binding affinity competition is expected to dictate hnRNP K activities. The observed changes in the sgRNA translation rate are likely not due to differences in the hnRNP K translocation rates, as both 181/25 and ΔhnRNPK-BS1 induced hnRNP K cytoplasmic translocation, the latter more so than the former strain, assumedly as a form of compensation for diminished hnRNPK–vRNA interaction. Changes in structural protein synthesis may thus be attributed to diminished hnRNPK–sgRNA interaction, as indicated by the coimmunoprecipitation and RT-qPCR experiments. Previous SINV studies illustrated that hnRNPK–vRNA disruption does not affect the recruitment of non-structural proteins to the replication initiation complex [71]. Therefore, hnRNP K is postulated to play a role in sgRNA translation via direct interaction with sgRNA, as previously demonstrated in other alphavirus studies [71]. However, the mechanisms behind such regulation remain unknown. One possibility is that hnRNP K performs sgRNA stabilization, thus mediating the avoidance of vRNA decay and inactivation. Dramatic decreases in both sgRNA and gRNA transcript levels in the ΔhnRNPK-BS1 mutants were initially assumed to be due to hnRNP K involvement in transcription regulation. However, such diminishments in transcript levels can be attributed to vRNA decay, as the downregulation of hnRNP K in the 181/25-infected astrocytes resulted in ablated vRNA transcript levels. Furthermore, the mere inactivation of nsP4 activity via sofosbuvir-mediated inhibition resulted in significantly lower transcript levels in the ΔhnRNPK-BS1 mutants versus those in 181/25, suggesting that the hnRNPK–vRNA interaction may serve a role in vRNA stabilization. However, dramatic changes in sgRNA, but not gRNA, translation indicate that hnRNP K more likely plays a role not only in the stabilization of sgRNA, but also via the interaction with the host translational machinery, such as elongation factor 1-alpha (EF-1α) and eukaryotic translation initiation factor 4A (eIF4E) [80,81,82,83]. Thus, further investigations into hnRNP K host and viral coimmunoprecipitants are needed to verify such postulated translational machinery interactions, as previously observed in retrovirus studies demonstrating hnRNP K acting as an internal ribosomal entry site (IRES) transactivating factor (ITAF) for HIV-1 vRNA translation [84].

Considering coimmunoprecipitation experiments illustrating hnRNP K propensity to interact with sgRNA, as specifically determined by the hnRNPK–vRNA interaction, it also remains possible that continuous hnRNP K association with viral sgRNA, including during sgRNA transport to the endoplasmic reticulum (ER), is necessary for effective sgRNA translation due to hnRNP K-mediated post-transcriptional stability, regulation, or even localization. Under ER stress conditions, as occur during alphavirus infections [85,86], the host-mediated shut-down of cap-dependent mRNA translation via protein kinase R-like endoplasmic reticulum kinase (PERK) phosphorylation of eukaryotic translation initiation factor 2A (eIF2A) occurs [87]. However, translation re-initiation to maintain ER homeostasis can occur via activating translated factor 4 (ATF4) [88,89]. Previous groups have illustrated the hnRNP K-mediated regulation of both eIF2A/ATF4 [90], as well as the hnRNP K inhibition of ribosomal 60S and 40S subunit conglomeration and the hnRNP K/hnRNP E1-mediated hindrance of 80S ribosomal assembly via RNP binding to the 3′UTR differentiation control element (DICE) and thus, eIF4F-independent translation shutoff [91]. Therefore, under ER stress conditions, hnRNP K may interact and modulate sgRNA translation, perhaps modulating the vRNA or sgRNA structure to enable scanning and thus, translation, by host ribosomes.

Host sequestration of translation initiation factors into stress granules is a known antiviral response in CHIKV-infected cells in vitro [86]. However, viral pathways initiated to prevent such sequestration have been observed within CHIKV. For example, the CHIKV nsP3 macrodomain (MD) exhibits stress granule formation inhibitory functions [86], while hnRNP K phosphorylation, a post-translationally modified form commonly found within SINV sgRNA-rich replicase machinery, is known to have disrupted the translational silencing capacity due to diminished DICE binding affinity [92,93]. Considering the ubiquitous negative modulatory roles of hnRNP K, it remains possible that the hnRNPK–vRNA-binding site shares homology with such host regulatory sequences, and thus acts as a competitive, nonfunctional RNA decoy or “sponge” with high hnRNP K binding affinity [94], thereby sequestering hnRNP K and diminishing its availability to regulate viral translation using host machinery such as eIF2A, eIF4F, and ATF4. Alternatively, hnRNPK–vRNA may act to sequester antiviral microRNAs (miRNAs) and competitive endogenous RNAs (ceRNAs), thereby reciprocally influencing and outcompeting host microRNA levels, a hypothesis known as “competitive viral and host RNAs,” or cvhRNAs [95]. Such alphavirus-mediated mRNA competition has previously been proposed [96] and may explain the diminished hnRNPK–vRNA-binding-mediated E2 glycoprotein translatability.

Dedication: This publication is dedicated to the late Dr. Diane E. Griffin, who unexpectedly passed during the final internal revisions of this manuscript. Dr. Griffin’s dedication to her colleagues and students, as well as her passion for acute encephalitic alphavirus research, will be remembered by all of those whom she has touched. We mourn the loss of a giant in the field of virology but are grateful for Dr. Griffin’s myriads of impactful contributions that will continue to be felt worldwide for generations to come.

## Figures and Tables

**Figure 1 viruses-16-01918-f001:**
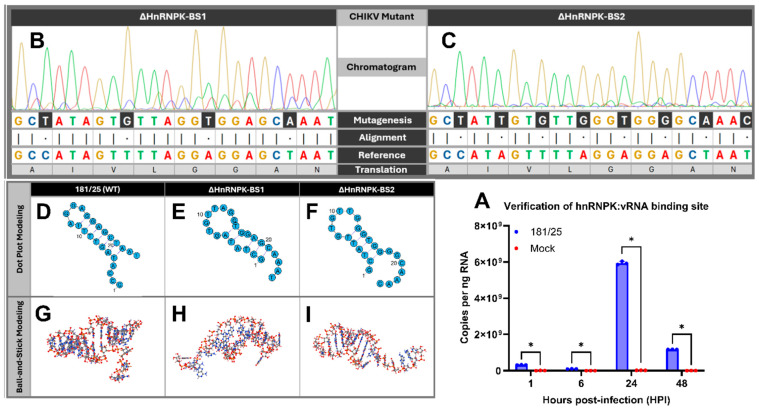
Mutagenesis of the heterogeneous ribonucleoprotein K (hnRNP K) chikungunya virus (CHIKV) viral RNA (vRNA)-binding site modifies the vRNA secondary structure. (**A**) RT-qPCR analysis of total RNA bound to coimmunoprecipitated hnRNP K protein collected from C8-D1A murine astrocytes infected with CHIKV 181/25 (WT) versus mock demonstrated hnRNP K interaction with the postulated hnRNPK–vRNA binding site originally identified via cross-sequence alignment with the Sindbis virus (SINV) hnRNPK–vRNA reference sequence. HnRNP K vRNA-binding site sequence annotations of attenuated CHIKV 181/25 versus (**B**) ΔhnRNPK-BS1 and (**C**) ΔhnRNPK-BS2 mutants with four or eight silent mutation substitutions, respectively. Site-directed silent mutations are highlighted in black and cross-aligned with the CHIKV 181/25 hnRNPK–vRNA-binding site consensus sequence. Translational products are delineated in a gray box below the alignments to illustrate the lack of change in amino acid translation. Predicted dot plot modeling of secondary structure of (**D**) CHIKV 181/25, (**E**) ΔhnRNPK-BS1, and (**F**) ΔhnRNPK-BS2 hnRNPK–vRNA-binding site sequences, as modeled using RNAComposer and RiboSketch software. Silent mutations resulted in RNA secondary structure modifications, as illustrated by the addition of an internal loop, an expanded hairpin loop, and disrupted overhang structures found in the vRNA of both ΔhnRNPK mutants. Three-dimensional ball-and-stick RNA secondary structure modeling schematics of the (**G**) CHIKV 181/25, (**H**) ΔhnRNPK-BS1, and (**I**) ΔhnRNPK-BS2 mutants, as generated by Geneious (Java Version 11.0.18+10) using the standard Corey-Pauling-Koltun color convention. Statistical significance was determined via two-way ANOVAs, assuming Gaussian distribution; * *p* < 0.05, mock versus 181/25.

**Figure 2 viruses-16-01918-f002:**
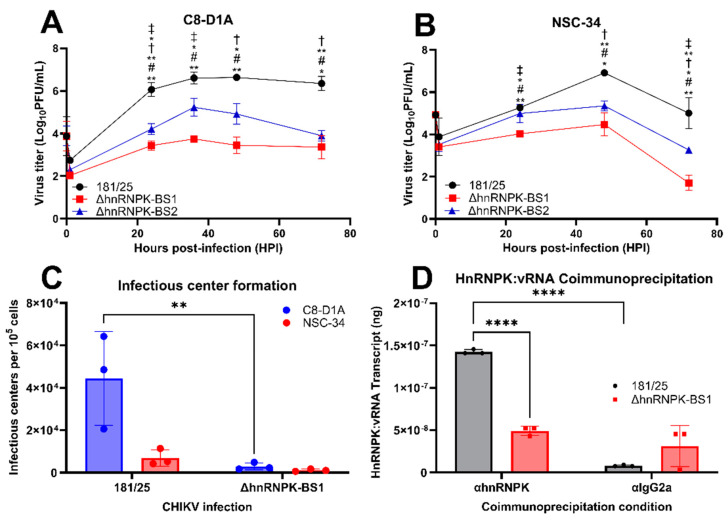
Mutagenesis of the heterogeneous ribonucleoprotein K CHIKV viral RNA-binding site negatively regulates viral replication and disrupts hnRNPK–vRNA binding. Murine astrocytic C8-D1A and neuronal NSC-34 cell lines were infected with the CHIKV 181/25, ΔhnRNPK-BS1, or ΔhnRNPK-BS2 passage 1 (P1) mutant virus at a multiplicity of infection (MOI) of 5. Supernatant was collected at various timepoints and viral titers in (**A**) murine C8-D1A astrocytes and (**B**) NSC-34 neurons were determined via Vero plaque assays; #, statistical difference from 181/25; †, significant difference from ΔhnRNPK-BS1; ‡, significant difference from ΔhnRNPK-BS2. (**C**) Mutagenesis of hnRNPK–vRNA diminishes CHIKV infectious center formation in C8-D1A astrocytes. (**D**) RT-qPCR analysis of total RNA pulled-down from coimmunoprecipitated hnRNP K protein isolated from CHIKV 181/25 or ΔhnRNPK-BS1-infected C8-D1A cells illustrated significantly diminished hnRNPK–vRNA-binding site transcript quantities bound to hnRNP K in ΔhnRNPK-BS1-infected cells. The experimental design included the use of three biological replicates per condition. Individual replicate values are denoted by symbols delineated by the legend, while standard deviations of the mean are represented by error bars. Statistical significance was determined via two-way ANOVAs, assuming Gaussian distribution; * *p* < 0.05; ** *p* < 0.01; **** *p* < 0.0001.

**Figure 3 viruses-16-01918-f003:**
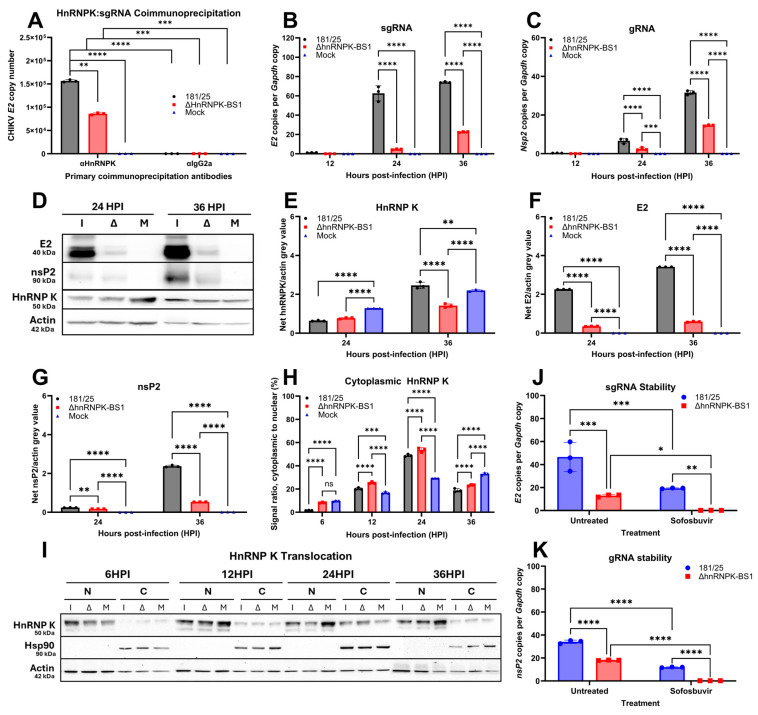
Heterogeneous ribonucleoprotein K interacts with CHIKV subgenomic RNA (sgRNA) and hnRNPK–vRNA binding regulates sgRNA translation. (**A**) Coimmunoprecipitation was performed using protein lysates harvested from murine astrocytes infected with CHIKV 181/25, CHIKV ΔhnRNPK-BS1, or media (mock) only. Protein pull-down was performed using αHnRNPK or αIgG2a isotype control and agarose-conjugated beads, followed by RNA isolation of protein-linked RNA, cDNA synthesis, and RT-qPCR to quantify sgRNA expression. RT-qPCR analysis of total RNA isolated from CHIKV 181/25 or ΔhnRNPK-BS1-infected C8-D1A astrocytes was performed to quantify (**B**) sgRNA and (**C**) genomic RNA (sgRNA) transcript copies, as normalized to host *Gapdh* copy numbers. (**D**) Western blot analysis illustrating E2 glycoprotein and nsP2 expression in CHIKV 181/25, ΔhnRNPK-BS1, or mock-infected C8-D1A cells. Densitometric analyses to quantify signal intensities of (**E**) hnRNP K, (**F**) E2, and (**G**) nsP2 from the blot depicted in (**D**). (**H**) Densitometric analyses of hnRNP K signal intensity in (**I**) immunoblotting depicting nuclear (N) and cytoplasmic (C) fractions of CHIKV 181/25, CHIKV ΔhnRNPK-BS1 (Δ), or mock (M)-infected C8-D1A astrocytes at various hours post-infection (HPI). Hsp90 was utilized as a cytoplasmic-specific marker. (**J**) C8-D1A cells were infected with CHIKV 181/25 or ΔhnRNPK-BS1 for 1 h, followed by 50 µM sofosbuvir treatment. Lysate was collected at 12 HPI, and sgRNA and (**K**) gRNA were quantified via RT-qPCR. The experimental design included the use of three biological replicates per condition. Individual replicate values are denoted by symbols delineated by the legend, while standard deviations of the mean are represented by error bars. Statistical significance was determined via two-way ANOVAs, assuming Gaussian distribution; * *p* < 0.05; ** *p* < 0.01; *** *p* < 0.001; **** *p* < 0.0001.

**Figure 4 viruses-16-01918-f004:**
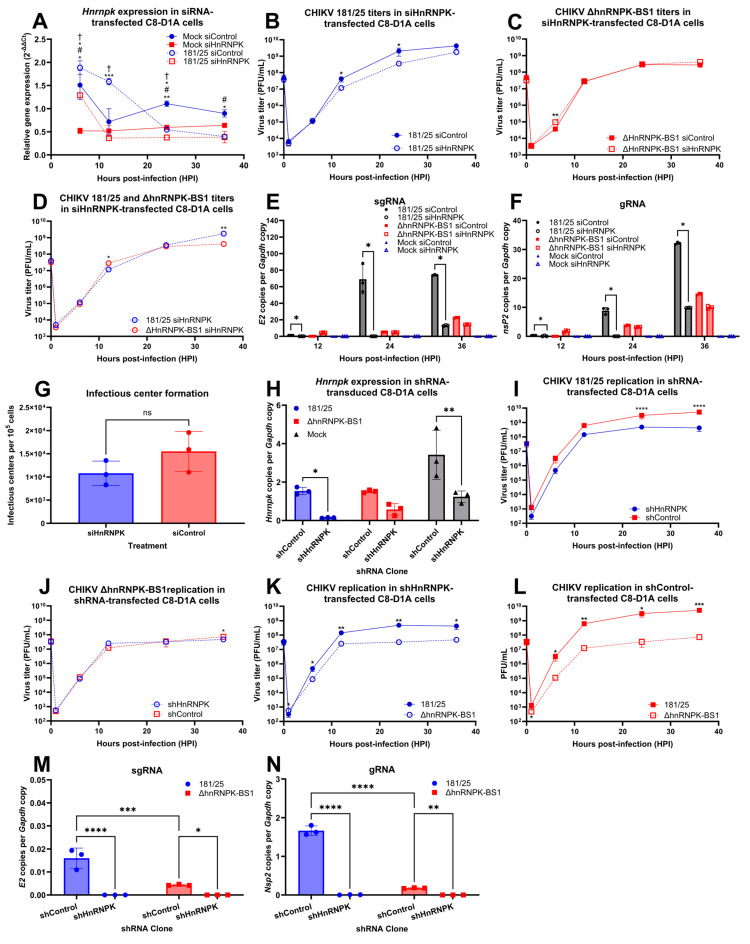
Host hnRNP K availability regulates CHIKV replication in astrocytes. (**A**) Small interfering RNA (siRNA)-mediated *Hnrnpk* gene knockdown was performed in murine astrocytes in vitro for 48 h prior to infection with CHIKV 181/25 at an MOI of 5. Quantification of *Hnrnpk* gene expression, as normalized to *Gapdh*-VIC, was then performed via RT-qPCR using RNA lysates harvested at designated timepoints. #, statistical difference between Mock siControl versus Mock siHnRNPK; †, statistical difference between 181/25 siControl versus 181/25 siHnRNPK. Vero plaque assays were used to quantify viral titers in longitudinal supernatants collected from siHnRNPK or siControl-transfected C8-D1A cells infected with (**B**) CHIKV 181/25 or (**C**) ΔhnRNPK-BS1. (**D**) CHIKV 181/25 or ΔhnRNPK-BS1 titers were additionally compared in siHnRNPK-transfected cells. RT-qPCR analysis of RNA isolated from siHnRNPK versus siControl-transfected, CHIKV 181/25-, ΔhnRNPK-BS1, or mock-infected C8-D1A cells was performed to quantify (**E**) relative sgRNA or (**F**) gRNA quantities. (**G**) Infectious center assays of siHnRNPK or siControl-transfected, CHIKV 181/25-infected C8-D1A cells were performed to quantify replication initiation. HnRNP K was constitutively knocked down in C8-D1A cells via short hairpin RNA (shRNA) vectors generating small interfering RNAs (siRNAs) that mediate hnRNP K gene expression interference. (**H**) Knockdown was confirmed via RT-qPCR analysis. Vero plaque assays demonstrating (**I**) CHIKV 181/25 and (**J**) ΔhnRNPK-BS1 titers in C8-D1A shHnRNPK versus shControl. Comparisons of CHIKV 181/25 versus ΔhnRNPK-BS1 titers in (**K**) shHnRNPK and (**L**) shControl-transduced C8-D1A cells. RT-qPCR analysis of lysates collected 24 HPI demonstrated significantly diminished (**M**) sgRNA (*E2*) and (**N**) gRNA (*Nsp2*) transcript levels in both CHIKV 181/25 and ΔhnRNPK-BS1-infected C8-D1A shHnRNPK clones. Experimental design included the use of three biological replicates per condition. Individual replicate values are denoted by symbols delineated by the legend, while standard deviations of the mean are represented by error bars. Statistical significance was determined via two-way ANOVAs, assuming Gaussian distribution; * *p* < 0.05; ** *p* < 0.01; *** *p* < 0.001; **** *p* < 0.0001; ns (non-significant).

**Figure 5 viruses-16-01918-f005:**
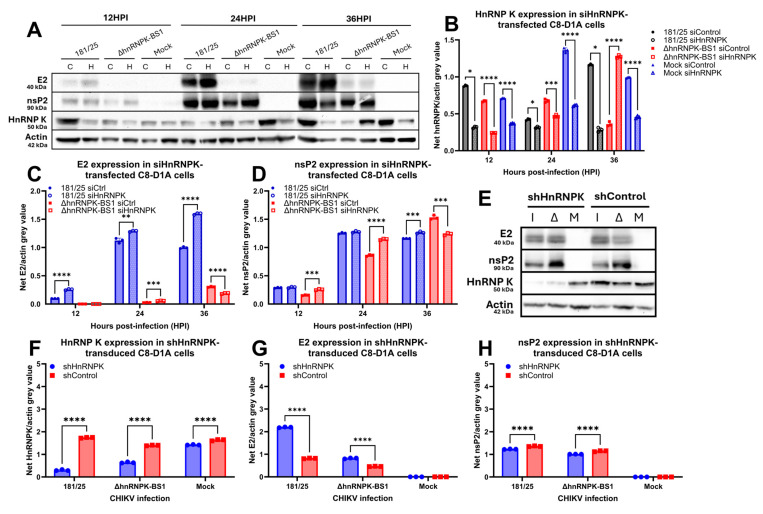
Altered CHIKV replication with respect to host hnRNP K availability regulates viral protein translation. (**A**) Western blot analysis of lysates collected from siHnRNPK-transfected astrocytes infected with CHIKV 181/25 versus ΔhnRNPK-BS1 virus illustrated dramatic differences in E2 glycoprotein expression. Densitometric analyses of the Western blot illustrated in (**A**) were conducted by quantifying the net mean pixelation intensity by determining mean gray values of (**B**) hnRNP K, (**C**) E2, and (**D**) nsP2 normalized to actin. (**E**) Protein lysates collected from CHIKV 181/25 or ΔhnRNPK-BS1-infected C8-D1A astrocytes with constituent hnRNP K knockdown via shRNA lentiviral construct transduction were collected at 24 HPI. Lysates were processed via Western blot analysis, and densitometry was performed to examine (**F**) hnRNP K, (**G**) E2, and (**H**) nsP2 expression. Experimental design included the use of three biological replicates per condition. Individual replicate values are denoted by symbols delineated by the legend, while standard deviations of the mean are represented by error bars. Statistical significance was determined via two-way ANOVAs, assuming Gaussian distribution; * *p* < 0.05; ** *p* < 0.01; *** *p* < 0.001; **** *p* < 0.0001.

**Table 1 viruses-16-01918-t001:** List of primer sequences utilized.

Primer Set	Sequence (5′→3′)
CHIKV *E2* 992 Forward *	GAA GAG TGG GTG ACG CAT AAG
CHIKV *E2* 1011 Reverse *	TGG ATA ACT GCG GCC AAT AC
CHIKV *E2* TaqMan Probe *	[FAM]-ATC AGG TTA ACC GTG CCG ACT GAA-[MGB NFQ]
CHIKV *Nsp2* 1247 Forward *	GTA CGG AAG GTA AAC TGG TAT GG
CHIKV *Nsp2* 1359 Reverse *	TCC ACC TCC CAC TCC TTA AT
CHIKV *Nsp2* TaqMan Probe *	[FAM]-TGCAGAACCCACCGAAAGGAAACT-[MGB NFQ]
*Hnrnpk* Forward	CGC TAT GAT GGC ATG GTT GG
*Hnrnpk* Reverse	AGA TCA CCA TAT GAG CCA CGG
HnRNPK–vRNA Forward	CGC CCC TTG TTG TCG AAG AT
HnRNPK–vRNA Reverse	GGT GGT GAC CTG GAA CAA AGA
ΔHnRNPK-BS1 SDM Forward	CAA GGG GCG CGT GGT GGC TAT AGT GTT AGG GGG AGC GAA TGA AGG AGC CCG TAC AG
ΔHnRNPK-BS1 SDM Reverse	CTG TAC GG GCT CCT TCA TTC GCT CCC CCT AAC ACT ATA GCC ACC ACG CGC CCC TTG
ΔHnRNPK-BS2 SDM Forward	CAA GGG GCG CGT GGT GGC TAT TGT GTT GGG TGG GGC AAA CGA AGG AGC CCG TAC AG
ΔHnRNPK-BS2 SDM Reverse	CTG TAC GGG CTC CTT CGT TTG CCC CAC CCA ACA CAA TAG CCA CCA CGC GCC CCT TG
HnRNPK–vRNA Sequencing	CGC CCC TTG TTG TCG AAG AT
mm.Ri-Hnrnpk.13.1 Forward	rGrGrA rArArG rArCrU rUrGrG rCrUrU rGrAr
mm.Ri-Hnrnpk.13.1 Reverse	rGrArA rArUrU rUrArU rCrArA rGrCrC rArArG
mm.Ri-Hnrnpk.13.2 Forward	rArArA rCrUrU rGrGrG rArUrU rCrUrG rCrArA
mm.Ri-Hnrnpk.13.2 Reverse	rGrUrG rUrCrA rArUrU rGrCrA rGrArA rUrCrC
mm.Ri-Hnrnpk.13.3 Forward	rArUrA rCrUrG rArGrU rArUrC rArGrU rGrCrU
mm.Ri-Hnrnpk.13.3 Reverse	rCrArA rUrArU rCrArG rCrArC rUrGrA rUrArC

* Primers delineated in [54].

**Table 2 viruses-16-01918-t002:** Antibody product list.

Target	Host/Isotype	Conjugate	Catalog Number	Company
Actin	Mouse IgG1k	Unconjugated	MAB1501	Millipore Sigma, Burlington, VT, USA
CHIKV E2 (Chk187)	Mouse IgG	Unconjugated	N/A	Gift from Prof. Michael Diamond, Washington University, St. Louis, MO, USA
HnRNP K	Mouse monoclonal IgG2ak	Unconjugated	sc-28380	Santa Cruz Biotechnology, Dallas, TX, USA
IgG2a isotype control	Mouse IgG2a	Unconjugated	MAB0031	Bio-Techne Corporation, Minneapolis, MN, USA
Mouse IgG [H] + [L] chains	Horse pAb	HRP	7076S	Cell Signaling Technology, Danvers, MA, USA
nsP2	Rabbit IgG	Unconjugated	GTX636897	GeneTex, Irvine, CA, USA
Rabbit IgG [H] + [L] chains	Goat pAb	HRP	7074S	Cell Signaling Technology

## Data Availability

Data not included in the manuscript will be made available from the primary authors on request.

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
