# Peer review of "Heterogeneous Ribonucleoprotein K Is a Host Regulatory Factor of Chikungunya Virus Replication in Astrocytes"

_viruses, 2024, doi:10.3390/v16121918_

Round 1

Reviewer 1 Report

Comments and Suggestions for Authors

1)      I suggest changing the title of the manuscript because neural cells do not include astrocytes that you also analyse in the study by replacing “neural” with "the central nervous system’s cells”

2)      The name of the virus appearing in the title should be written in capital letters, i.e. Chikungunya virus.

3)       Explain the low level of Fetal Bovine Serum (FBS) used in culture media to maintain culture cells used in the study and verify the amount of FBS % given. The recommended amount is 10%.

4)       Add a space bar character before the bracket of each number of literature citations throughout the manuscript.

Author Response

Comment 1: “I suggest changing the title of the manuscript because neural cells do not include astrocytes that you also analyse in the study by replacing “neural” with the central nervous system’s cells”.

Response 1: Agreed. We have changed the title to the following: “Heterogeneous Ribonucleoprotein K is a Host Regulatory Factor of Chikungunya Virus Replication in Astrocytes.” Following our previous Viruses publication formatting (Kim et al., Viruses, 2022), we capitalized all letters and removed any reference to “neural” cells. Since our publication predominantly focuses on CHIKV host factor interactions within astrocytes, we changed the title to reflect this. In addition we made changes to Line 552 and Line 560 to be more specific to astrocyte.

Comment 2: “The name of the virus appearing in the title should be written in capital letters, i.e. Chikungunya virus.”

Response 2: Modified. Please refer to “Response 1.”

Comment 3: “Explain the low level of Fetal Bovine Serum (FBS) used in culture media to maintain culture cells used in the study and verify the amount of FBS % given. The recommended amount is 10%.”

Response 3: We utilize media with lower FBS only during infection procedures. Our lab has optimized CHIKV inoculation and has historically found enhanced virus-cell interaction, and thus infection efficiency, at lower FBS concentrations. However, we use 10% FBS whilst maintaining cells; 1%, thus, is only used during virus inoculations.

Comment 4: “Add a space bar character before the bracket of each number of literature citations throughout the manuscript.”

Response 4: We have added a space bar character before the bracket of each citation throughout the manuscript.

Reviewer 2 Report

Comments and Suggestions for Authors

In this manuscript, Heterogeneous ribonucleoprotein K (hnRNP 16K) serves as a crucial RNA binding host factor that regulates CHIKV replication through modulation of subgenomic RNA translation. Generally speaking, this article is quite meaningful. However, the work need to be revised:

1. Why chose the hnRNP 16K? And among various members of the heterogeneous nuclear ribonucleoproteins (hnRNPs) subfamily, why just focused on hnRNP 16K?

2. In the figures, please label the molecular weights of all proteins.

3. Figure 3D, why is there no expression of the in mock-infected C8-D1A cells.

4. In addition, the manuscript uses a variety of cells for experiments, which is suggested to be noted in the picture. Otherwise, for example in Fig 5A, cells cannot be distinguished.

Author Response

Reviewer 2

Comment 1: “Why chose the hnRNP 16K? And among various members of the heterogeneous nuclear ribonucleoproteins (hnRNPs) subfamily, why just focused on hnRNP 16K?”

Response 1: Previous studies have demonstrated hnRNP K acting as a host factor in the prototype alphavirus, Sindbis virus (SINV) (Burnham et al., 2007). Similarly, others have identified heterogeneous ribonucleoprotein (hnRNP) M, K, and I as host factors involved with SINV replication (LaPointe et al., 2018). We initially performed Western blotting experiments to examine hnRNP expression in chikungunya virus (CHIKV)- and mock-infected NSC-34 neurons and C8-D1A astrocytes. However, our preliminary experiments only demonstrated differential hnRNP K expression in astrocytes, but not neurons. Furthermore, no changes in hnRNP I or hnRNP M were observed in either CHIKV-infected or uninfected astrocytes versus neurons. Therefore, our immunoblotting experiments indicated that hnRNP K would be an interesting host protein to investigate due to its proclivity to showcase cell type-dependent expression modification during CHIKV infection.

Comment 2: “In the figures, please label the molecular weights of all proteins.”

Response 2: We have labeled all immunoblotting images with respective molecular weights of each protein investigated.

Comment 3: “Figure 3D, why is there no expression of the in mock-infected C8-D1A cells.”

Response 3: Non-structural protein 2 (nsP2) and envelope 2 (E2) glycoprotein are both viral proteins. Mock-infected cells were not infected with virus, but rather virus-free media.

Comment 4: “In addition, the manuscript uses a variety of cells for experiments, which is suggested to be noted in the picture. Otherwise, for example in Fig 5A, cells cannot be distinguished.”

Response 4: The manuscript makes use of both NSC-34 neurons and C8-D1A astrocytes only in Figure 2. The remainder of the manuscript focuses on wild-type C8-D1A astrocytes with transient hnRNP K knock-down or constitutive hnRNP K knock-down. Therefore, figures 3-5 only make use of C8-D1A astrocytes, as delineated in figure descriptions.